# Excitonic States in GaAs/Al*_x_*Ga_1−_*_x_*As Quantum Wells: Direct Coulomb Interaction Modeling via Finite Element Electrostatics and Parametric Analysis Under Impurity and Field Effects

**DOI:** 10.3390/nano15171345

**Published:** 2025-09-01

**Authors:** Fabian Andres Castaño, David Laroze, Carlos Alberto Duque

**Affiliations:** 1Scientific Instrumentation and Microelectronics Research Group-GICM, Physics Institute, Exact and Natural Sciences Faculty, Universidad de Antioquia UdeA, Calle 70 No. 52-21, Medellín 050010, Colombia; fabian.castano@udea.edu.co; 2Bioinstrumentation and Clinical Engineering Research Group-GIBIC, Bioengineering Department, Engineering Faculty, Universidad de Antioquia UdeA, Calle 70 No. 52-21, Medellín 050010, Colombia; 3Instituto de Alta Investigación, Universidad de Tarapacá, Casilla 7D, Arica 1000000, Chile; dlarozen@academicos.uta.cl; 4Grupo de Materia Condensada-UdeA, Instituto de Física, Facultad de Ciencias Exactas y Naturales, Universidad de Antioquia UdeA, Calle 70 No. 52-21, Medellín 050010, Colombia

**Keywords:** exciton, quantum well, binding energy, shallow impurities, finite element method, COMSOL-Multiphysics, electric field, magnetic field, Coulomb interaction

## Abstract

This study presents a comprehensive numerical investigation of excitonic states in GaAs quantum wells embedded in ${\mathrm{Al}}_{x}$${\mathrm{Ga}}_{1-x}$As barriers, incorporating the effects of donor and acceptor impurities, external electric and magnetic fields, and varying well widths. The electron and hole wavefunctions are computed by directly solving the Schrödinger equation using the finite element method in cylindrical coordinates, without assuming trial forms. To evaluate the exciton binding energy, the implementation and comparison of two independent approaches were performed: a numerical integration method based on elliptic function corrections, and a novel finite element electrostatic formulation using COMSOL Multiphysics v5.6. The latter computes the Coulomb interaction by solving Poisson’s equation with the hole charge distribution and integrating the resulting potential over the electron density. Both methods agree within 1% and capture the spatial and field-induced modifications in excitonic properties. The results show that quantum confinement enhances binding in narrow wells, while donor impurities and electric fields reduce binding via spatial separation of carriers. Magnetic fields counteract this effect by providing radial confinement. The FEM-based electrostatic method demonstrates high spatial accuracy, computational efficiency, and flexibility for complex heterostructures, making it a promising tool for exciton modeling in low-dimensional systems.

## 1. Introduction

Excitons in semiconductor quantum wells (QWs) remain a central topic in condensed matter physics due to their enhanced binding energies in reduced dimensions, which often exceed bulk values and enable stable excitonic states even at room temperature [1,2,3]. In recent years, renewed attention has been given to excitonic effects in QWs owing to their role in strong light–matter coupling and polaritonic devices, as well as their potential applications in optoelectronics and quantum information technologies. For instance, recent studies have reported novel excitonic resonances in wide GaAs/AlGaAs heterostructures and their tunability by external fields [4,5], as well as the observation of room-temperature exciton–polaritons in engineered QW structures [6,7]. These developments illustrate that excitons in semiconductor QWs remain not only of fundamental interest but also central to current device-oriented research, further motivating the need for accurate computational methods such as those proposed in this work.

Theoretical efforts to model excitons in QWs span from variational approaches to advanced numerical techniques. Variational methods have been widely adopted due to their conceptual simplicity and low computational cost [2,3,6,8,9]; however, their precision is critically dependent on the choice of trial wavefunctions. Although refinements such as composite or correlated propositions improve performance [3,6,10,11,12], these approaches struggle to incorporate structural imperfections, inhomogeneous potentials, or localized impurity effects.

Alternative modeling techniques aim to address these limitations. Finite-difference and finite-element schemes allow for direct discretization of the Schrödinger equation, capturing the confinement and composition-dependent features without assuming functional forms [9,13,14,15,16]. Fractional-dimensional models [17,18,19,20] introduce semiempirical dimensionality to mimic confinement effects, albeit with reduced physical interpretability. Path-Integral Monte Carlo (PIMC) techniques provide highly accurate results that include many-body correlations and disorder effects [4,20,21], but remain computationally prohibitive for extensive parametric studies. Other methods, such as those of Leavitt and Little [16,22,23], offer integration-based simplifications that trade accuracy for efficiency.

A critical survey of the recent literature reveals persistent gaps. First, the explicit inclusion of donor or acceptor impurities—especially as spatially localized Coulombic centers—remains rare, despite their ubiquitous presence in real nanostructures [16,21,24,25,26]. Second, most works employ quasi-1D or 2D approximations that neglect the full 3D character of QWs and the role of asymmetries [14,27,28,29]. Third, few studies investigate the full range of well widths, especially in the ultra-thin regime (e.g., <10 nm), where quantum confinement and impurity effects are strongly non-linear. Recent studies in low-dimensional perovskite systems [15,30,31,32,33] further highlight the importance of accurate exciton modeling under confinement and disorder, motivating the extension of robust computational techniques to diverse material platforms.

To address these challenges, we develop a direct numerical framework based on the finite element method (FEM) in cylindrical coordinates, tailored for axisymmetric quantum wells. This approach resolves the electron and hole wavefunctions without any trial assumptions and includes the effects of quantum confinement, spatially localized donor impurities, and externally applied static electric and magnetic fields. A key novelty of this work is the comparison of two independent computational methods to evaluate the excitonic Coulomb interaction: (1) a numerical quadrature approach using elliptic function corrections implemented in C/OpenMP, and (2) a fully self-consistent electrostatic formulation using the FEM electrostatics module in COMSOL-Multiphysics. The latter enables the direct computation of Coulomb coupling from charge distributions derived from the hole wavefunction, offering enhanced precision and scalability.

This article presents a comprehensive evaluation of excitonic states in ${\mathrm{GaAs/Al}}_{x}$${\mathrm{Ga}}_{1-x}$As quantum wells, covering a range of well widths, impurity types, and external field configurations. The results highlight the interplay between spatial localization and field-induced separation, demonstrating that the FEM-based electrostatic method in COMSOL-Multiphysics provides an efficient and accurate tool for computing Coulomb interactions in complex excitonic systems.

This paper is organized as follows. Section 2 presents the theoretical background and describes the underlying theory. Section 3 presents the methodology. Section 4 presents the results. Section 5 presents a discussion of our main findings and, finally, Section 6 outlines our main conclusions.

## 2. Theoretical Background

Understanding excitonic states in semiconductor heterostructures requires a robust theoretical framework that simultaneously captures the effects of single-particle confinement and the Coulomb interaction between electrons and holes. This is especially challenging in systems like GaAs QWs embedded in ${\mathrm{Al}}_{x}$${\mathrm{Ga}}_{1-x}$As barriers, where both quantum confinement and impurity effects can significantly alter excitonic behavior.

### 2.1. Effective Mass Approximation and Confinement Model with Impurity Interaction

Within the effective mass approximation, the behavior of charge carriers (electrons and holes) confined in a QW is governed by a modified Schrödinger equation. Near the edge of the conduction or valence band, the dispersion relation is approximated as parabolic, allowing the replacement of the actual mass by an effective mass $m^{*}$, which accounts for the effects of the band structure. An external static magnetic field $\vec{B}=B_{0}\;\hat{z}$ and a static electric field $\vec{E}=E_{0}\;\hat{z}$ are considered, both oriented along the growth axis of the QW. To incorporate these fields into the quantum description, the canonical momentum $\vec{p}=-i\;\hbar \;\nabla$ is replaced according to the minimal coupling prescription:(1)$$\vec{p}\to \vec{p}+q_{e/h}\;\vec{A},$$
where $q_{e}=-e$ is for electrons, $q_{h}=+e$ is for holes, and $\vec{A}$ is the vector potential associated with $\vec{B}$. Here, the symmetric gauge is adopted in cylindrical coordinates (ρ,ϕ,z), where the vector potential $\vec{A}$ is expressed by:(2)$$\vec{A}=\frac{B_{0}}{2}\rho \;\hat{\phi }.$$

The static electric field $\vec{E}$ is incorporated through the potential energy term $-q_{e/h}\;E_{0}z$, leading to a linear potential along the growth direction. The total Hamiltonian, which includes the magnetic and electric fields, the confinement potential $V_{e/h}\left(\vec{r}\right)$, and an impurity potential $V_{\mathrm{imp}}\left(\vec{r}\right)$ modeling a donor or acceptor impurity located at ρ=0 and $z=z_{\mathrm{imp}}$ (typically at mid-well), is given by:(3)$$H_{e/h}=\frac{1}{2\;m_{e/h}^{*}}{\left(\vec{p}+q_{e/h}\;\vec{A}\right)}^{2}+V_{e/h}\left(\vec{r}\right)+V_{\mathrm{imp}}\left(\vec{r}\right)-q_{e/h}\;E_{0}z.$$

Here, $V_{e/h}\left(\vec{r}\right)$ describes the spatial confinement of electrons and holes, and $V_{\mathrm{imp}}\left(\vec{r}\right)$ represents the Coulomb interaction between the carriers and the impurity located at ${\vec{r}}_{\mathrm{imp}}$. These potentials, together with the effective mass approximation and the inclusion of external fields, define the behavior of charge carriers in the quantum well.(4)$$V_{e/h}\left(\vec{r}\right)=\left\{\begin{array}{cc} 0, & \mathrm{inside}\;\mathrm{the}\;\mathrm{GaAs}\;\mathrm{quantum}\;\mathrm{well},\\ V_{0}^{e/h}, & \mathrm{in}\;\mathrm{the}\;{\mathrm{Al}}_{0.3}{\mathrm{Ga}}_{0.7}\mathrm{As}\;\mathrm{barriers}, \end{array}\right.$$(5)$$V_{\mathrm{imp}}\left(\vec{r}\right)=n_{\mathrm{imp}}\;\frac{q_{e/h}\;\left|e\right|}{4\;\pi \;\varepsilon_{0}\;\varepsilon_{r}\;\left|\vec{r}-{\vec{r}}_{\mathrm{imp}}\right|},$$
where $n_{\mathrm{imp}}$ is the concentration of impurities, $\varepsilon_{0}$ is the vacuum permittivity, and $\varepsilon_{r}$ is the relative dielectric constant of GaAs. For donor impurities, $n_{\mathrm{imp}}=1$ is used, while for acceptor impurities, $n_{\mathrm{imp}}=-1$ is applied. The impurity potential is modeled as a Coulomb potential centered on the impurity position ${\vec{r}}_{\mathrm{imp}}$.

To numerically solve the Schrödinger equation, the system is modeled in cylindrical coordinates (r,ϕ,z), leveraging the axial symmetry of the problem. The full time-independent Schrödinger equation for the electron or hole wavefunction $\psi_{e/h}\left(\vec{r}\right)$ is:(6)$$H_{e/h}\psi_{e/h}\left(\vec{r}\right)=E_{e/h}\psi_{e/h}\left(\vec{r}\right),$$
which, after expanding the Hamiltonian, becomes as shown in:(7)$$\begin{array}{c} \left(-\frac{\hbar^{2}}{2\;m_{e/h}^{*}}\nabla^{2}+\frac{q_{e/h}^{2}\;B_{0}^{2}}{8\;m_{e/h}^{*}}\rho^{2}-q_{e/h}\;E_{0}\;z+V_{e/h}\left(\vec{r}\right)+V_{\mathrm{imp}}\left(\vec{r}\right)\right)\;\psi_{e/h}\left(\vec{r}\right)\\ =E_{e/h}\;\psi_{e/h}\left(\vec{r}\right). \end{array}$$

The following material parameters were used [19]: effective mass of the electron $m_{e}^{*}=0.067\;m_{0}$ and effective mass of the heavy hole $m_{h}^{*}=0.600\;m_{0}$ in ${\mathrm{Al}}_{x}$${\mathrm{Ga}}_{1-x}$As barrier domains; effective mass of the electron $m_{e}^{*}=0.093\;m_{0}$ and effective mass of the heavy hole $m_{h}^{*}=0.510\;m_{0}$ in GaAs quantum well domain; electron rest mass $m_{0}=9.1094\times 10^{-31}\;\mathrm{kg}$; conduction band offset $V_{0}^{e}=300\;\mathrm{meV}$ and valence band offset $V_{0}^{h}=177\;\mathrm{meV}$; the magnetic induction magnitude $B_{0}=\left[0,5,10,20\right]\;T$; the electric field $E_{0}=\left[0,0.01,0.1,1\right]\;kV/cm$; and the relative dielectric constant of GaAs $\varepsilon_{r}=12.9$.

Dirichlet boundary conditions ($\psi (\vec{r}\to \infty )=0$) are applied on the outer boundaries to emulate an open structure and prevent unphysical reflections. These infinite domains absorb the decaying parts of the wavefunction, replicating the behavior of an infinite medium surrounding the quantum well.

The donor impurity is located at the position ${\vec{r}}_{\mathrm{imp}}=\left(0,z_{\mathrm{imp}}\right)$, where $z_{\mathrm{imp}}$ corresponds to the center of the GaAs QW for each case of parametric sweep.

This framework allows us to study both the “pure” exciton state (when $V_{\mathrm{imp}}\left(\rho ,z\right)=0$) and the exciton in the presence of a donor impurity (with $V_{\mathrm{imp}}\left(\rho ,z\right)$ active), as well as the influence of external static fields. This comprehensive model provides insight into the interplay of quantum confinement, impurity effects, and external perturbations on excitonic properties.

### 2.2. Schrödinger Equation in Cylindrical Coordinates (ρ,ϕ,z) with External Fields

Since this system exhibits axial symmetry, we can work in cylindrical coordinates (ρ,ϕ,z). The solution of the Schrödinger equation for electrons and holes confined in a QW under the influence of an external magnetic field and an electric field can be expressed as a wavefunction of separated variables. The wavefunction $\psi_{e/h}\left(\rho ,\phi ,z\right)$ can be expressed as a product of radial, azimuthal, and axial components, i.e.:(8)$$\psi_{e/h}\left(\rho ,\phi ,z\right)=R_{e/h}\left(\rho ,z\right)\;e^{i\;m\;\phi },$$
where $R_{e/h}\left(\rho ,z\right)$ is the radial and axial part of the wavefunction, and $e^{i\;m\;\phi }$ captures the azimuthal dependence, with *m* being the magnetic quantum number or the azimuthal quantum number. The Hamiltonian in cylindrical coordinates can be expressed as:(9)$$\begin{array}{cc} H_{e/h}= & -\frac{\hbar^{2}}{2\;m_{e/h}^{*}}\left(\frac{\partial^{2}}{\partial \rho^{2}}+\frac{1}{\rho }\frac{\partial }{\partial \rho }+\frac{\partial^{2}}{\partial z^{2}}-\frac{m^{2}}{\rho^{2}}\right)+\frac{q_{e/h}^{2}\;B_{0}^{2}}{8\;m_{e/h}^{*}}\;\rho^{2}\\ \; & +\frac{\hbar \;|q_{e/h}|\;B_{0}}{2m_{e/h}^{*}}\;m-q_{e/h}\;E_{0}\;z+V_{e/h}\left(\rho ,z\right)+V_{\mathrm{imp}}\left(\rho ,z\right). \end{array}$$

This form allows for solving the problem in two dimensions: the radial and axial coordinates (ρ,z), with the ϕ-dependence fully captured by the $e^{i\;m\;\phi }$ factor and the corresponding centrifugal and magnetic angular momentum terms.

## 3. Methodology

This section presents the methodological procedure for evaluating the full excitonic state in a QW formed by a GaAs layer between two ${\mathrm{Al}}_{x}$${\mathrm{Ga}}_{1-x}$As barriers. The geometry of interest is introduced, followed by the Coulomb interaction model, which is solved using a four-dimensional Coulomb integral approach under axial symmetry and the FEM implemented in COMSOL-Multiphysics v5.6. The Schrödinger equations for the electron and hole were solved using the Partial Differential Equations (PDEs) Module with the Eigenvalue Solver, while the Coulomb interaction was computed with the Electrostatics Module by solving Poisson’s equation for the electron and hole charge densities.

### 3.1. Geometry of Interest

The simulation domain is defined in COMSOL-Multiphysics using axisymmetric geometry, as shown in Figure 1. The structure comprises two rectangular ${\mathrm{Al}}_{x}$${\mathrm{Ga}}_{1-x}$As barriers and a central GaAs QW. Specifically, the first ${\mathrm{Al}}_{x}$${\mathrm{Ga}}_{1-x}$As barrier extends from z=0 to z=10 nm and from ρ=0 to ρ=98 nm. The second barrier has the same dimensions and is placed symmetrically along the *z*-axis, sandwiching the GaAs QW. In all simulations, the GaAs quantum well was surrounded by ${\mathrm{Al}}_{x}$${\mathrm{Ga}}_{1-x}$As barriers of 10 nm on each side, ensuring sufficient confinement and negligible tunneling through the boundaries. In addition, the overall FEM domain was extended with a buffer region up to the artificial infinity boundary, so that the electron and hole wavefunctions decayed to numerical zero before reaching the external limits where the infinity conditions were imposed.

To analyze excitonic states under different conditions, we simulate three cases: (i) no impurity, (ii) donor impurity, and (iii) acceptor impurity. The point impurity is placed at ρ=0, in the center of the GaAs layer, along the *z*−axis. This setup allows for direct comparison of binding energies and wavefunction localization in pristine and impurity-influenced scenarios.

The thickness of the GaAs QW is varied parametrically from 20 nm to 100 nm in 5 nm steps, and from 100 nm to 200 nm in 10 nm steps. Infinite element boundary conditions are applied to the bottom, top, and right boundaries to mimic an infinite surrounding medium, as was shown in Figure 1. Dirichlet conditions $\psi_{e/h}\left(\vec{r}\to \infty \right)=0$ are enforced at the domain limits to ensure proper decay of wavefunctions.

A uniform square mesh with a maximum element size of 0.005 nm is used throughout the domain. The mesh ensures high spatial resolution and numerical convergence, even for narrow well widths. COMSOL-Multiphysics’s adaptive meshing ensures high mesh quality, with skewness approaching 100%.

### 3.2. Coulomb Interaction and Exciton Formation

Once the electron and hole wavefunctions and their corresponding energy levels are obtained from separate Schrödinger equation simulations in COMSOL-Multiphysics, solving the PDE equations, the Coulomb interaction energy *J* is computed as follows:(10)$$J=\int_{V_{e}}\int_{V_{h}}\frac{e^{2}}{4\;\pi \;\varepsilon_{0}\;\varepsilon_{r}}\frac{|\psi_{e}\left({\vec{r}}_{e}\right){|}^{2}{\left|\;\psi_{h}\left({\vec{r}}_{h}\right)\right|}^{2}}{|{\vec{r}}_{e}-{\vec{r}}_{h}|}\;dV_{h}\;dV_{e},$$
where $e=1.6022\times 10^{-19}\;C$ is the elementary charge; $\varepsilon_{0}$ is the vacuum permittivity; and $\psi_{e}\left({\vec{r}}_{e}\right)$, $\psi_{h}\left({\vec{r}}_{h}\right)$ are the spatial wavefunctions of the electron and hole, respectively. This double integral accounts for the spatial probability distribution of the charge carriers and quantifies their mutual Coulomb interaction over the entire domain. The calculation is carried out using either COMSOL-Multiphysics’s built-in integration operators or through post-processing of the exported wavefunctions. The wavefunctions $\psi_{e}\left({\vec{r}}_{e}\right)$ and $\psi_{h}\left({\vec{r}}_{h}\right)$ are exported from COMSOL-Multiphysics and processed using a parallelized C program (OpenMP). This computation is repeated across all QW widths. The source code is publicly available at https://github.com/fabioc9675/CoulombExcitonBinding_prj.git (accessed on 24 May 2025).

To simplify the calculation of the Coulomb integral *J*, the axial symmetry of the system is considered. This allows the original six-dimensional integral to be reduced to a four-dimensional form, reducing the computational complexity from $O(N^{6})$ to $O(N^{4})$, where *N* is the number of discretized mesh points in each dimension of the wavefunctions. This reduction in dimensionality is possible by analytically integrating the angular variable using elliptic functions, as shown in the following equation.(11)$$J=\frac{e^{2}}{4\;\pi \;\varepsilon_{0}\;\varepsilon_{r}}\int_{S_{e}}\int_{S_{h}}|R_{e}\left(\rho_{e},z_{e}\right){|}^{2}\;{\left|R_{h}\left(\rho_{h},z_{h}\right)\right|}^{2}\;V_{K}\left(r_{p}\right)\;dS_{h}\;dS_{e},$$
where $dS_{h}=\rho_{h}\;d\rho_{h}\;dz_{h}$, $dS_{e}=\rho_{e}\;d\rho_{e}\;dz_{e}$, and the elliptic potential correction is given by:(12)$$V_{K}\left(r_{p}\right)=\frac{8\;\pi \;K\left(\frac{r_{p}}{1+r_{p}}\right)}{r\;\sqrt{1+r_{p}}},$$
with $r_{p}=\frac{4\;\rho_{e}\;\rho_{h}}{r^{2}}$ and $r=\sqrt{{\left(\rho_{e}-\rho_{h}\right)}^{2}+{\left(z_{e}-z_{h}\right)}^{2}}$. Here K(k) is the complete elliptic integral of the first kind. To avoid costly evaluations of special functions, a rational polynomial approximation of K(k) was used, i.e.:(13)$$K\left(k\right)\approx A_{K}\left(P_{K}\right)-B_{K}\left(P_{K}\right)\ln\left(P_{K}\right),$$
where $P_{K}=1-k^{2}$, and $A_{K}$, $B_{K}$ are polynomials in $P_{K}$, evaluated using Horner’s method (Equation (Equation 14)) [19]; also, the coefficients $a_{i}$ and $b_{i}$ are defined in Table 1:(14)$$\begin{array}{cc} A_{K}\left(P_{K}\right) & =\left(\left(\left(a_{0}P_{K}+a_{1}\right)P_{K}+a_{2}\right)P_{K}+a_{3}\right)P_{K}+a_{4},\\ B_{K}\left(P_{K}\right) & =\left(\left(\left(b_{0}P_{K}+b_{1}\right)P_{K}+b_{2}\right)P_{K}+b_{3}\right)P_{K}+b_{4}. \end{array}$$

This approach ensures high numerical precision for k∈[0,0.99] while greatly improving computational efficiency.

### 3.3. Coulomb Interaction Using FEM in COMSOL

As an independent validation, the Coulomb interaction *J* using a FEM in COMSOL-Multiphysics was computed. First, the Schrödinger equations for the electron and hole were solved to obtain $\psi_{e}\left(\rho ,\phi ,z\right)$ and $\psi_{h}\left(\rho ,\phi ,z\right)$, respectively. The hole charge density is defined as follows:(15)$$\Omega_{h}\left(\rho ,\phi ,z\right)=e\;{\left|\psi_{h}\left(\rho ,\phi ,z\right)\right|}^{2},$$

This charge density is used as the source term in the electrostatics module to solve the Poisson equation:(16)$$\nabla \;\cdot \;\left(\varepsilon \left(\vec{r}\right)\;\nabla V_{h}\left(\vec{r}\right)\right)=-\Omega_{h}\left(\vec{r}\right),$$
where $\varepsilon (\vec{r})$ is the spatially dependent electric permittivity of the quantum well, $V_{h}\left(\vec{r}\right)$ is the electric potential produced by the hole charge distribution, and $\Omega_{h}\left(\vec{r}\right)$ is the hole charge density. Infinite boundary domains are handled using Dirichlet conditions such that $V_{h}\to 0$ as $|\vec{r}|\to \infty$.

On the other hand, the electron charge density is defined as(17)$$\Omega_{e}\left(\rho ,\phi ,z\right)=e\;{\left|\psi_{e}\left(\rho ,\phi ,z\right)\right|}^{2}.$$

Finally, the Coulomb interaction energy *J* is evaluated in COMSOL as:(18)$$\begin{array}{cc} J & =\int_{V}\Omega_{e}\left(\vec{r}\right)\;V_{h}\left(\vec{r}\right)\;dV\\ J & ={\;\int \int \int \;}_{V}\left(e\;|\psi_{e}{\left(\rho ,\phi ,z\right)|}^{2}\right)\;V_{h}\left(\rho ,\phi ,z\right)\;\rho \;d\rho \;d\phi \;dz. \end{array}$$

This FEM-based approach avoids explicit multidimensional integration and leverages COMSOL’s solver capabilities to compute *J* directly from charge distributions. The results are compared with the integral-based method for cross-validation. A parametric sweep of well thickness reveals the dependence of excitonic binding strength on quantum confinement.

Although the central impurity at ρ=0 is used for clarity, the FEM approach also supports off-axis or extended impurity distributions by placing the corresponding source terms at arbitrary positions. Point-like sources are mildly regularized and the mesh is refined locally to ensure convergence.

## 4. Results

This section presents the results obtained from the study of exciton states in a GaAs QW confined between two ${\mathrm{Al}}_{x}$${\mathrm{Ga}}_{1-x}$As barriers. The energy levels of the individual charge carriers and the resulting excitonic state are analyzed under impurity-free and impurity-influenced conditions. Specifically, we examine the effect of QW width on the confinement energy, the spatial localization of the carriers, and the influence of external electric and magnetic fields, based on the parameters established in Section 2.1. These results are derived from the numerical solution of the Schrödinger equation for the electron and hole wavefunctions.

### 4.1. Solution of the Schrödinger Equation for the Electron and the Hole

The axisymmetric form of the time-independent Schrödinger equation was solved independently for the electron and hole in a GaAs QW embedded between two ${\mathrm{Al}}_{x}$${\mathrm{Ga}}_{1-x}$As barriers. Three impurity scenarios were considered: (i) no impurity, (ii) a donor impurity located at the center of the well, and (iii) an acceptor impurity at the same position. Each case was simulated under four different external conditions: (a) no applied fields, (b) an electric field along the *z*-direction, (c) a magnetic field along the *z*-axis, and (d) simultaneous application of both fields.

To illustrate the combined effects, Figure 2 shows the spatial distribution of the wavefunctions for both the hole and the electron under each impurity configuration in the presence of an electric field $E_{0}=1\;kV/cm$ and a magnetic field $B_{0}=10\;T$. In Figure 2, the well width is set to 80 nm to enhance the visibility of field-induced effects. Although this width is relatively large, similar effects could be observed in narrower wells by increasing the field intensity (e.g., $E_{0}∼100\;kV/cm$), as is shown in Figure 3, where the width is reduced to 8 nm in order to highlight the influence of impurities and external fields in a narrow quantum well.

As illustrated in Figure 2 and Figure 3, both the presence of external fields and the type of impurity significantly influence the spatial distribution of the wavefunctions. The applied electric field, oriented along the *z*-axis, exerts a force in the opposite direction on the negatively charged electron, shifting its wavefunction toward the bottom of the well. In contrast, it exerts a force in the same direction on the positively charged hole, displacing its wavefunction toward the top of the well. In addition, the magnetic field introduces a radial confinement effect on both particles, drawing their wavefunctions toward the center of the quantum well. The color scale represents the probability density $|\psi \left(\vec{r}\right){|}^{2}$, with red indicating maximum probability and decreasing gradually towards cooler and lighter colors.

In addition to electric field effects, impurities introduce attractive or repulsive interactions depending on their nature (donor or acceptor) and the type of charge carrier (electron or hole). A donor impurity (positively charged) attracts the electron, concentrating its wavefunction at the center of the well, as shown in Figure 2a and Figure 3a. In contrast, it repels the hole, moving its wavefunction away from the center and inducing a slight deformation, as observed in Figure 2b and Figure 3b. Similarly, an acceptor impurity (negatively charged) attracts the hole, causing its wavefunction to localize near the center of the well (Figure 2f and Figure 3f), while it repels the electron, shifting its wavefunction outward and producing a mild distortion near the center, as shown in Figure 2e and Figure 3e.

It is also worth noting that the hole exhibits a more delocalized, free-like behavior, while the electron remains more tightly bound within the QW. This difference arises primarily from the disparity in effective masses: in GaAs, the hole has a significantly larger effective mass than the electron. As a result, its effective Bohr radius is much smaller—approximately 1.54 nm for acceptor-type impurities (holes), compared to about 10.3 nm for donor-type impurities (electrons). Consequently, the electron wavefunction is more spatially extended and more strongly influenced by the confinement potential, making it less responsive to external perturbations. In contrast, the hole wavefunction, being more localized, is more easily displaced by applied electric or magnetic fields. This distinction is clearly illustrated by comparing the unperturbed cases in Figure 2c,d and Figure 3c,d, where the electron remains more confined, while the hole distribution is broader and more susceptible to external influence.

### 4.2. Validation of the Binding Energy Calculation Using C/OpenMP and COMSOL

As previously described, the exciton binding energy was calculated using two approaches: a numerical integration method based on the elliptic function implemented in parallelized C code with OpenMP (Section 3.2), and a finite element method (FEM) implemented in COMSOL Multiphysics (Section 3.3).

To validate the FEM-based approach, the results obtained from both methods were compared for the Coulomb integral *J*. This integral was evaluated for a GaAs QW confined between two ${\mathrm{Al}}_{x}$${\mathrm{Ga}}_{1-x}$As barriers, with a donor impurity placed at the center of the well. The system was subjected to an external electric field of magnitude $E_{0}=1\;kV/cm$ and a magnetic field of $B_{0}=20\;T$. A sweep was performed over the width of the well from 10 nm to 200 nm. The results of the Coulomb integral *J* computed with both methods are shown in Figure 4.

As shown in Figure 4, both methods produce highly consistent results. In both cases, a peak binding energy close to 8.0 meV is observed for a well width of 10 nm. The binding energy remains nearly constant between 10 nm and 35 nm, which is probably due to the magnetic field-induced confinement of the carriers near the center of the well. Beyond this well width, the binding energy decreases progressively as the well width increases. This decline is attributed to the enhanced spatial separation between the electron and the hole caused by the combined effect of the electric field and the central impurity, which reduces the Coulomb interaction. For a well width of 200 nm, the binding energy drops to approximately 1.8 meV. The computational performance metrics for both methods are summarized in Table 2.

### 4.3. Exciton Binding Energy in the GaAs Quantum Well

To investigate the exciton behavior, several configurations were considered, including the presence or absence of impurities, electric fields, and magnetic fields. For each of these scenarios, the exciton binding energy was calculated and analyzed. Figure 5 shows the computed binding energy *J* for the case without impurities ($q_{\mathrm{imp}}=0$), evaluated under various combinations of electric and magnetic field strengths.

As illustrated in Figure 5, the binding energy decreases as the QW width increases. This reduction is due to the greater spatial volume available for the electron and hole wavefunctions to spread out, leading to a weaker Coulomb interaction between them. Moreover, the applied electric field exerts opposite forces on the electron and hole wavefunctions along the *z*-axis, pushing the electron/hole downwards/upwards. This field-induced spatial separation further reduces the binding energy. This effect becomes more pronounced as the electric field strength increases, as can be seen in the progressive panels of Figure 5 and in Figure 2. In particular, Figure 5d highlights how, for strong electric fields (e.g., $E_{0}=1\;kV/cm$), the binding energy approaches zero for well widths beyond 100 nm, indicating near-complete spatial separation of the excitonic pair. In contrast, the magnetic field has an opposing effect. As the magnetic field strength increases, it improves the radial confinement of both the electron and hole wavefunctions toward the center of the well (i.e., toward ρ=0), promoting stronger localization and overlap between the particles. This increases the binding energy, which counteracts the dissociating influence of the electric field.

The second case analyzed corresponds to a QW containing a donor impurity located in the center of the GaAs well ($q_{\mathrm{imp}}=+e$). As in the previous case, several combinations of electric and magnetic field intensities were evaluated. Figure 6 presents the computed exciton binding energies under these conditions.

As shown in Figure 6, the presence of the donor impurity significantly reduces the exciton binding energy due to its repulsive effect on the hole wavefunction and attractive interaction with the electron. This effect is evident in the absence of external fields, where the binding energy remains below 2.0 meV across all well widths. The application of a magnetic field partially counteracts this reduction by confining both wavefunctions toward the center of the well (ρ=0), thereby increasing their spatial overlap and enhancing the binding energy. However, the electric field still acts to separate the electron and hole along the *z*-axis, which limits the gain in binding energy as the well width increases.

In the third case, we evaluated the presence of an acceptor impurity ($q_{\mathrm{imp}}=-e$) located in the center of the GaAs QW. As in the previous cases, various combinations of electric and magnetic field intensities were considered. Figure 7 presents the calculated exciton binding energies under these conditions.

As shown in Figure 7, and in comparison with the donor impurity case in Figure 6, the presence of an acceptor impurity exerts an attractive force on the electron and a repulsive force on the hole. This results in a relatively low binding energy when no electric or magnetic fields are applied. However, as the magnetic field intensity increases, the binding energy improves as a result of enhanced confinement of both wavefunctions toward the center of the well. It should be noted that, in this configuration, the exciton binding energy tends to be higher than in the donor impurity case. This is primarily because the hole is less tightly bound to the impurity potential and can respond more freely to the magnetic field, increasing the wavefunction overlap and thus enhancing the binding energy.

Figure 8 provides a global comparison of the exciton binding energy under various conditions, including the presence or absence of impurities, as well as the influence of electric and magnetic fields. As observed, the magnetic field has the most significant impact among all factors considered. Its strong confinement effect pulls both electron and hole wavefunctions toward the center of the well, resulting in a substantial increase in the binding energy.

### 4.4. Scalar Electric Potential Field Generated by the Hole

As discussed in Section 3.3, the scalar electric potential generated by the hole can be evaluated by treating the hole wavefunction, multiplied by the elementary charge, as a volumetric charge distribution (see Equation (Equation 15)). From this charge distribution, it is possible to compute the resulting scalar electric potential field. Figure 9 presents a comparison matrix showing the electric potential field generated by the hole under different conditions, in the absence of an impurity, and the influence of external electric and magnetic fields. The results illustrate how the magnetic field enhances confinement toward the center of the quantum well, while the electric field causes a displacement of the potential in the *z*−direction.

## 5. Discussion

This study presents a detailed numerical analysis of excitonic states in GaAs/${\mathrm{Al}}_{x}$${\mathrm{Ga}}_{1-x}$As QWs, accounting for the effects of donor and acceptor impurities, external electric and magnetic fields, and well width. This approach combines a direct solution of the single-particle Schrödinger equation with two independent methods to compute the excitonic Coulomb interaction: a numerical integration method based on elliptic corrections and a finite element electrostatic formulation. This dual methodology provides both accuracy and cross-validation, and offers insight into the advantages and limitations of each computational strategy.

The numerical integration method implemented in parallelized C/OpenMP relies on the evaluation of a four-dimensional Coulomb integral with elliptic corrections, exploiting the system’s cylindrical symmetry. This method offers good accuracy and is suitable for extensive parametric sweeps; however, it requires careful treatment of singularities and substantial computational resources as the mesh resolution increases.

In contrast, the FEM implemented in COMSOL-Multiphysics leverages the electrostatics module to evaluate the scalar potential generated by the hole’s charge distribution. This field is then used to compute the interaction energy with the electron density via a direct spatial integration. The FEM-based approach is advantageous in its ability to treat arbitrary boundary conditions, complex geometries, and varying dielectric profiles without explicit multidimensional integration. The results of both methods agree within 1%, validating the precision of the COMSOL-based strategy. Moreover, COMSOL achieves faster runtimes and higher spatial resolution, especially for narrow wells. To the best of our knowledge, this is the first time this electrostatic formulation has been applied to excitonic systems in QWs, providing a flexible and scalable alternative to traditional integration techniques [3,17,22,34].

The two approaches used to compute the Coulomb interaction are complementary. The numerical quadrature with elliptic corrections is straightforward, transparent, and well-suited for systematic parameter sweeps. However, it is more challenging to extend to complex geometries, varying permittivity, or multiple off-axis impurities, and it requires careful handling of singularities. In contrast, the FEM-based electrostatic formulation solves Poisson’s equation directly from charge densities, making it naturally adaptable to arbitrary geometries, non-uniform permittivity, and realistic boundary conditions. Its main limitations are the dependence on mesh quality and boundary placement, as well as the need to regularize point-like sources. In our implementation, the electrostatic step is solved self-consistently for the given densities, whereas the Schrödinger equations for the electron and hole are treated independently. Both methods yield consistent results (within ∼1%). We recommend the quadrature method for simple, homogeneous systems and the FEM approach for cases where asymmetry or material complexity is important.

The simulations reveal that donor and acceptor impurities significantly alter the spatial distribution of the electron and hole wavefunctions, as shown in Figure 2. Donor impurities attract holes and repel electrons, leading to spatial separation and reduced wavefunction overlap, while acceptor impurities have the opposite effect Figure 8. This asymmetry results in a notable reduction in the exciton binding energy, especially for donor impurities.

Electric fields applied along the growth direction introduce a field-induced Stark effect: electrons are attracted toward the bottom of the well, while holes are repelled upward, further decreasing their spatial overlap and Coulomb interaction. This effect becomes dominant at field strengths above 10 kV/cm, where exciton dissociation becomes likely in wider wells.

Magnetic fields, on the other hand, provide radial confinement as a result of the Lorentz force, concentrating the charge densities toward the center of the well. This enhances wavefunction overlap and increases binding energy, partially compensating for the spatial separation induced by the electric field or impurities. The competition between electric and magnetic field effects creates a tunable regime in which exciton properties can be modulated [3,22].

Off-axis impurities break the cylindrical symmetry and localize carriers around the impurity site, which generally reduces electron–hole overlap and binding energy. Distributed impurities add potential fluctuations that further weaken binding on average. These scenarios are consistent with the trends reported here and can be naturally treated within the FEM–electrostatics framework.

The binding energy exhibits a non-monotonic dependence on well width. For widths below 20 nm, quantum confinement dominates, leading to strong quantization and improved Coulomb interaction [1,14]. As the well becomes wider, the electron and hole wavefunctions spread, reducing spatial overlap and weakening excitonic binding. This trend is further modulated by external fields and the presence of impurities [13,18,29,35].

Interestingly, in wells wider than 100 nm, a magnetic field of $B_{0}>5$ T partially restores the overlap by inducing radial localization. This behavior aligns with predictions from magneto-exciton theory and highlights the importance of considering both radial and axial confinement effects in wide quantum wells [2,21,36,37,38].

While this method captures key physical effects with high spatial resolution and generality, it does not include explicit electron–hole correlation in the Schrödinger solution, as the wavefunctions are computed independently. As a result, the absolute binding energy values may be underestimated when compared to variational models that incorporate correlated trial wavefunctions [3,22]. In the present GaAs/${\mathrm{Al}}_{x}$${\mathrm{Ga}}_{1-x}$As QWs, however, this underestimation remains small: comparison with the parallelized quadrature method shows deviations below 1%, confirming that the proposed approach is quantitatively reliable for the parameter space studied. However, the qualitative trends are consistent with theoretical expectations and experimental data, validating the method for comparative and parametric studies.

Another simplification in the present model is the use of constant effective masses and band offsets for GaAs/${\mathrm{Al}}_{x}$${\mathrm{Ga}}_{1-x}$As quantum wells. This assumption is commonly adopted for low-temperature, strain-free systems, where variations with respect to well width or external fields are negligible [1,16]. In real structures, however, these parameters can vary with temperature, strain, or alloy composition. Such changes would primarily shift the absolute values of the exciton binding energy, but the qualitative trends reported here, such as the dependence on well width, the reduction due to impurities and electric fields, and the partial recovery under magnetic confinement, remain robust. A more detailed treatment including parameter variability would be an interesting extension of this work, particularly for strained or high-temperature regimes.

The proposed FEM-based Coulomb calculation method opens new avenues for modeling more complex systems, such as coupled QWs, core–shell nanostructures, and optoelectronic devices under strain or non-uniform fields. Future work may extend the framework to include full two-body wavefunction solutions, self-consistent Schrödinger–Poisson models, time-dependent dynamics, and explicit dependencies on temperature, strain, or composition, as well as the effects of exciton–phonon coupling and many-body interactions in excited states.

## 6. Conclusions

This work presented a comprehensive numerical investigation of excitonic states in GaAs/${\mathrm{Al}}_{x}$${\mathrm{Ga}}_{1-x}$As quantum wells, considering the influence of donor and acceptor impurities, external electric and magnetic fields, and varying QW widths. The exciton binding energy was computed using two complementary approaches: a parallelized numerical integration scheme based on elliptic corrections and a finite element formulation implemented in COMSOL-Multiphysics. Both methods yielded consistent results within a 1% margin of error, confirming the accuracy and reliability of the FEM-based electrostatic model. The COMSOL approach demonstrated superior spatial resolution and computational efficiency, especially in narrow wells, and constitutes a novel application of electrostatics-based modeling to excitonic systems.

The study revealed that the type and location of impurities play a decisive role in determining the spatial configuration of the electron and hole wavefunctions. Donor impurities, which attract electrons and repel holes, lead to spatial separation and significantly reduced exciton binding energies. In contrast, acceptor impurities promote the localization of the hole near the center of the well, resulting in enhanced excitonic coupling. The application of external fields further modulates these effects.

Electric fields along the growth direction produce a Stark effect that displaces the electron and hole in opposite directions along the *z*-axis. This field-induced spatial separation leads to a pronounced decrease in binding energy, especially in wells wider than 100 nm, where exciton dissociation becomes likely. Magnetic fields counteract this effect by radially confining the charge carriers toward the center of the well, thereby increasing wavefunction overlap and partially recovering the binding energy.

A non-monotonic dependence of the exciton binding energy on the well width was observed. In narrow wells, strong confinement enhances the Coulomb interaction, resulting in higher binding energies. As the well width increases, the excitonic pair becomes more delocalized, resulting in reduced spatial overlap and weakened binding. However, for wide wells, the application of strong magnetic fields ($B_{0}>5$ T) can partially restore radial localization and increase the binding energy, in agreement with magneto-exciton theory.

The study also highlighted the asymmetry in the response of electrons and holes to external perturbations, attributed to their differing effective masses. While electrons remain more confined and less sensitive to field variations, holes exhibit broader spatial distributions and greater displacement under external fields, particularly in the presence of impurities.

Overall, the proposed modeling framework offers a robust and scalable platform for analyzing excitonic properties in QWs under realistic physical conditions. The electrostatic FEM-based method is particularly well-suited for extension to more complex systems, including coupled wells, strained heterostructures, and time-dependent phenomena. Future work may incorporate correlated two-particle wavefunctions and exciton–phonon interactions to capture finer details of excitonic dynamics and optical response.

## Figures and Tables

**Figure 1 nanomaterials-15-01345-f001:**
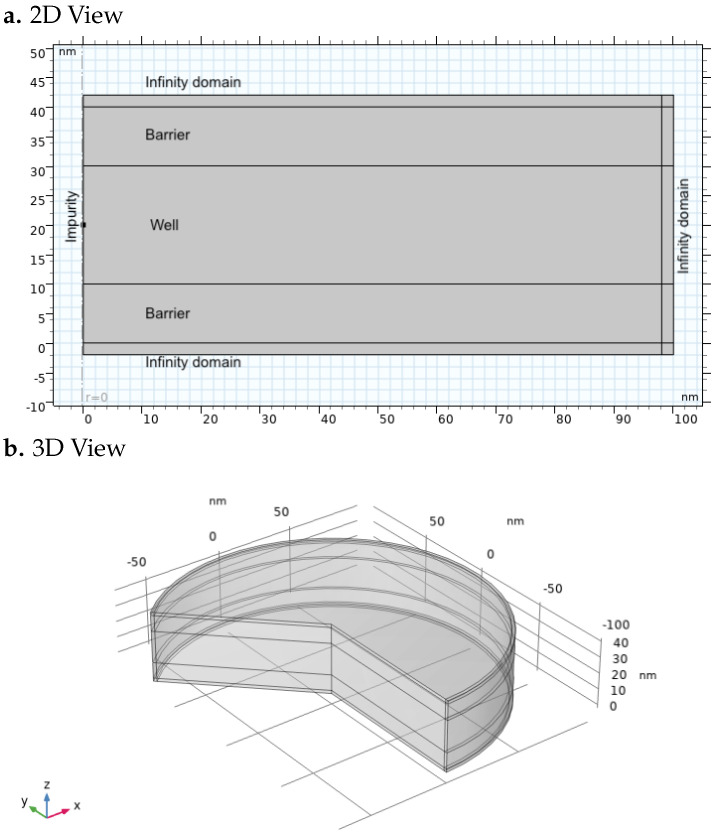
(**a**) The 2D distribution and (**b**) 3D visualization of the geometry.

**Figure 2 nanomaterials-15-01345-f002:**
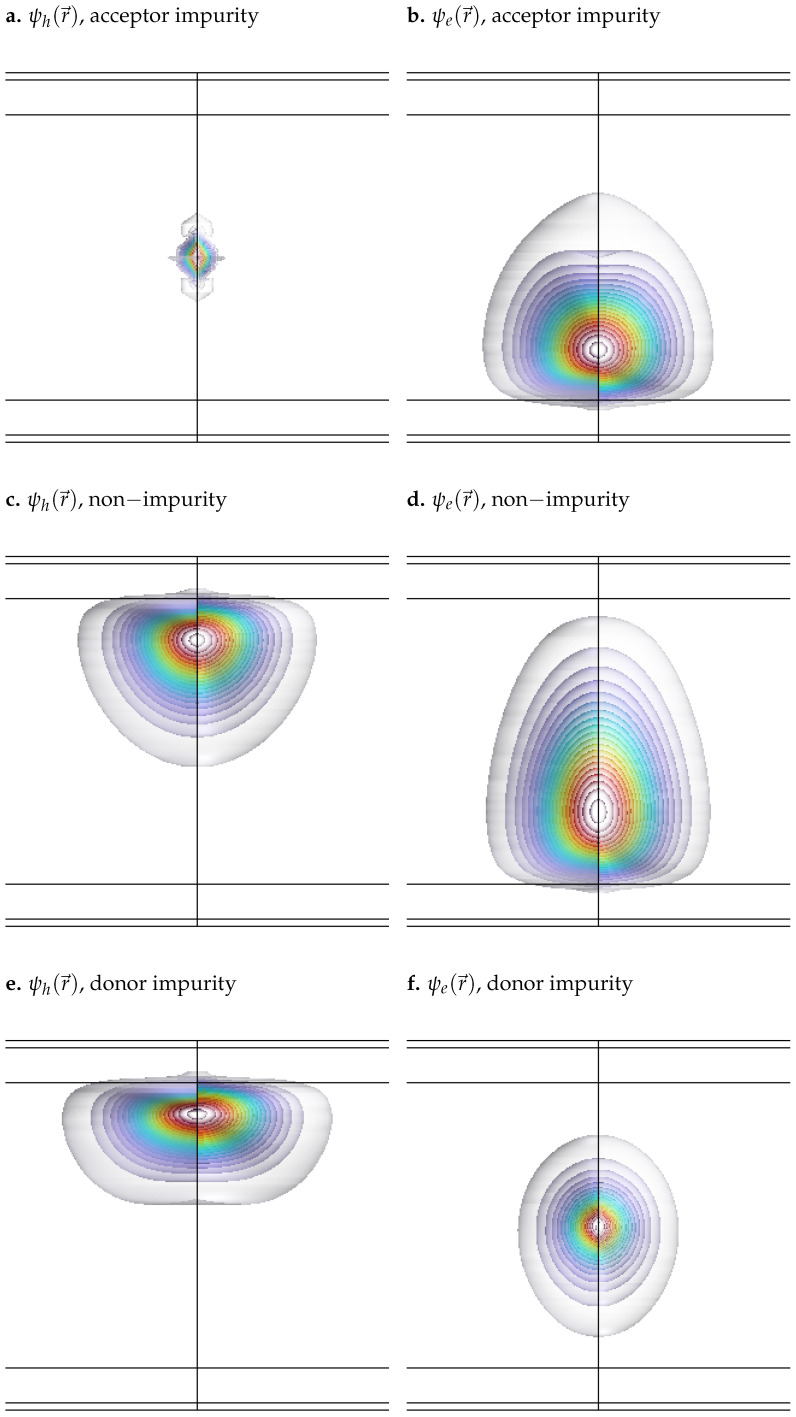
Electron and hole wavefunctions in a GaAs quantum well with ${\mathrm{Al}}_{x}$${\mathrm{Ga}}_{1-x}$As barriers under different impurity configurations. All simulations correspond to a well width of 80 nm, with applied electric and magnetic fields of $E_{0}=1\;kV/cm$ and $B_{0}=10\;T$, respectively. (**a**) Hole wavefunction with acceptor impurity. (**b**) Electron wavefunction with acceptor impurity. (**c**) Hole wavefunction without impurity. (**d**) Electron wavefunction without impurity. (**e**) Hole wavefunction with donor impurity. (**f**) Electron wavefunction with donor impurity.

**Figure 3 nanomaterials-15-01345-f003:**
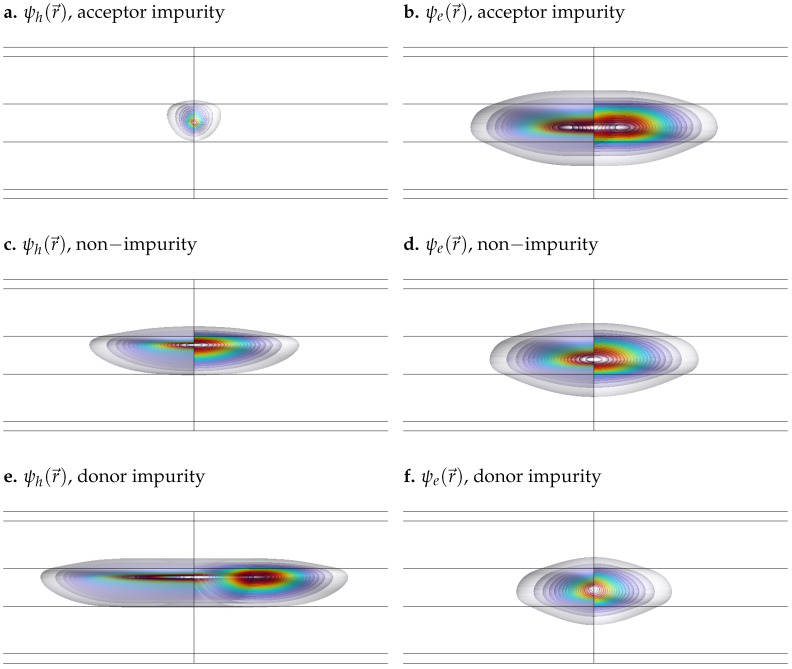
Electron and hole wavefunctions in a GaAs quantum well with ${\mathrm{Al}}_{x}$${\mathrm{Ga}}_{1-x}$As barriers under different impurity configurations. All simulations correspond to a well width of 8 nm, with applied electric and magnetic fields of $E_{0}=100\;kV/cm$ and $B_{0}=20\;T$, respectively. (**a**) Hole wavefunction with acceptor impurity. (**b**) Electron wavefunction with acceptor impurity. (**c**) Hole wavefunction without impurity. (**d**) Electron wavefunction without impurity. (**e**) Hole wavefunction with donor impurity. (**f**) Electron wavefunction with donor impurity.

**Figure 4 nanomaterials-15-01345-f004:**
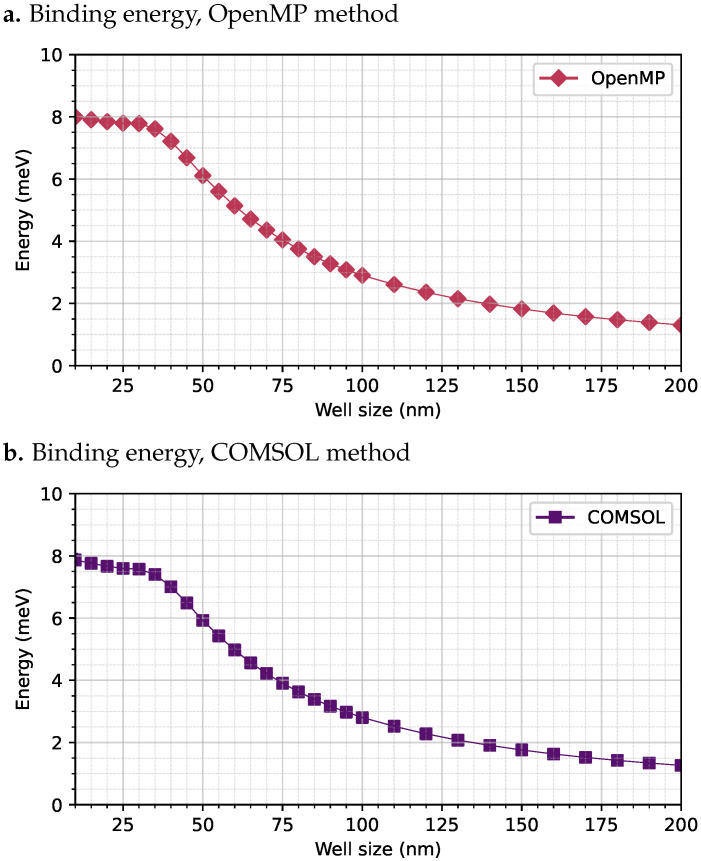
Comparison of the exciton binding energy *J* computed using the numerical integration method based on elliptic functions with parallel C/OpenMP code (**a**), and using the finite element method (FEM) implemented in COMSOL Multiphysics (**b**).

**Figure 5 nanomaterials-15-01345-f005:**
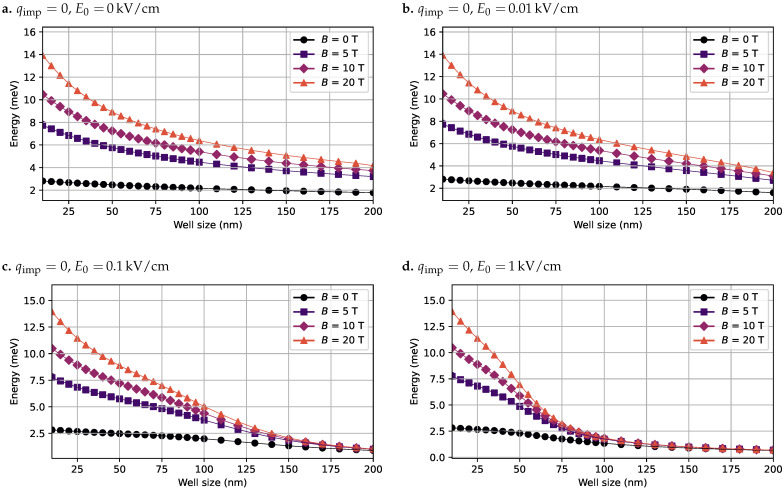
Exciton binding energy calculated in the absence of impurities ($q_{\mathrm{imp}}=0$) for various combinations of electric and magnetic field strengths. The quantum well width is swept from 10 nm to 200 nm. (**a**) Binding energy curves with no electric field. (**b**) Binding energy curves with an electric field of 0.01 kV/cm. (**c**) Binding energy curves with an electric field of 0.1kV/cm. (**d**) Binding energy curves with an electric field of 1kV/cm. In all cases, the magnetic field was swept over $B_{0}=\left[0,5,10,20\right]\;T$.

**Figure 6 nanomaterials-15-01345-f006:**
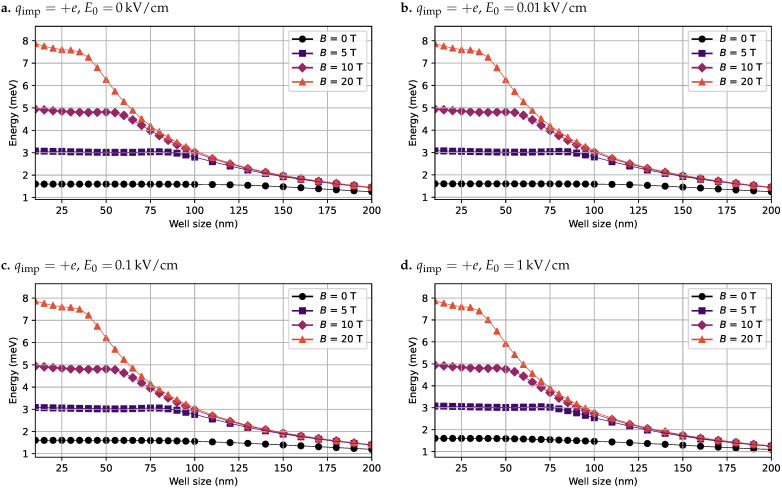
Exciton binding energy computed in the presence of a donor impurity located at the center of the GaAs quantum well, for various combinations of electric and magnetic field intensities. The well width is swept from 10 nm to 200 nm. (**a**) Binding energy curves with no applied electric field. (**b**) Binding energy curves with an electric field of 0.01
kV/cm. (**c**) Binding energy curves with an electric field of 0.1
kV/cm. (**d**) Binding energy curves with an electric field of 1 kV/cm. In all cases, the magnetic field is swept over $B_{0}=\left[0,5,10,20\right]\;T$.

**Figure 7 nanomaterials-15-01345-f007:**
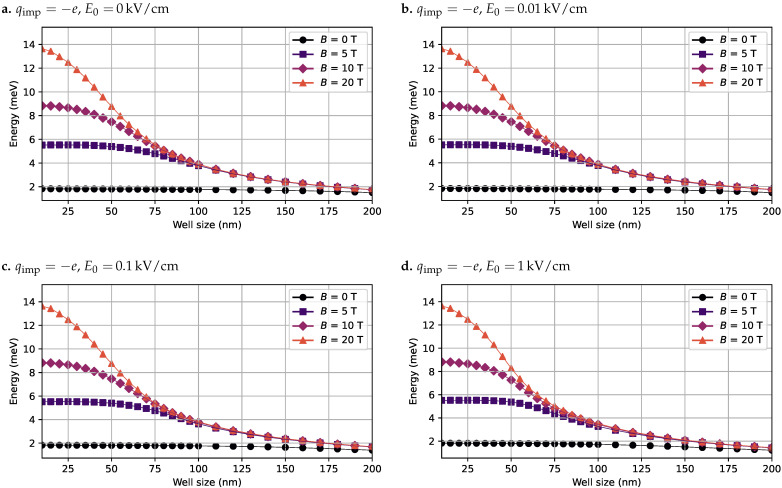
Exciton binding energy computed in the presence of an acceptor impurity located at the center of the GaAs quantum well, for various combinations of electric and magnetic field intensities. The well width is swept from 10 nm to 200 nm. (**a**) Binding energy curves with no applied electric field. (**b**) Binding energy curves with an electric field of 0.01
kV/cm. (**c**) Binding energy curves with an electric field of 0.1
kV/cm. (**d**) Binding energy curves with an electric field of 1 kV/cm. In all cases, the magnetic field is swept over $B_{0}=\left[0,5,10,20\right]\;T$.

**Figure 8 nanomaterials-15-01345-f008:**
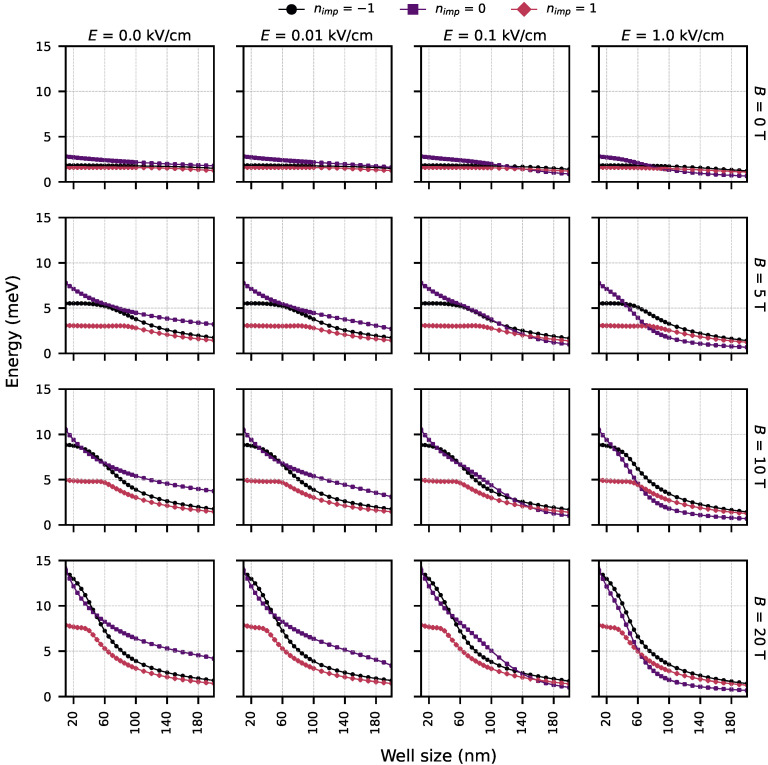
Matrix of exciton binding energy under different conditions, comparing the effects of electric and magnetic fields and the presence or absence of impurities. All subplots share the same scale on both axes: binding energy (meV) and well width (nm).

**Figure 9 nanomaterials-15-01345-f009:**
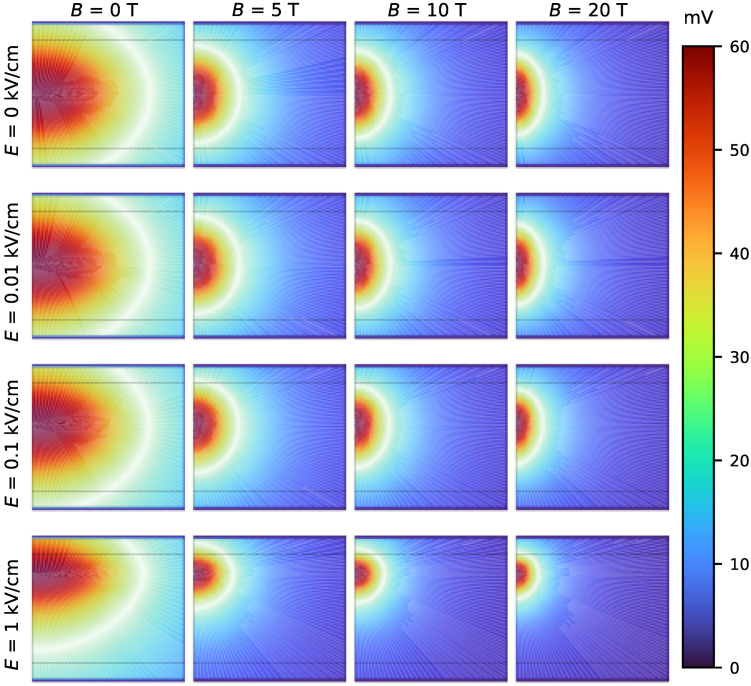
Scalar electric potential field generated by the volumetric charge distribution associated with the hole wavefunction in a 80 nm well width in absence of impurity. The results show increased confinement due to the magnetic field and displacement along the *z*−axis induced by the external electric field.

**Table 1 nanomaterials-15-01345-t001:** Coefficients used in the rational approximation of the complete elliptic integral of the first kind, K(k).

Index	$a_{i}$	$b_{i}$
0	0.01451196212	0.00441787012
1	0.03742563713	0.03328355346
2	0.03590092383	0.06880248576
3	0.09666344259	0.12498593597
4	1.38629436112	0.50000000000

**Table 2 nanomaterials-15-01345-t002:** Comparison between the parallelized (C/OpenMP) method and COMSOL Multiphysics for Coulomb integral evaluation.

Parameter	C/OpenMP	COMSOL FEM
Number of CPU cores	8	8
Number of mesh nodes	11,269	15,151
Mesh quality (skewness)	100%	100%
Total number of integrals	37	37
Average time per integral	12.37 s	2.35 s
Total execution time	81.11 s	10.86 s
Relative error in *J*	Reference	<1%
Spatial resolution	1 nm	0.2 nm

## Data Availability

The data presented in this study are available on the public repository https://github.com/fabioc9675/CoulombExcitonBinding_prj.git (accessed on 24 May 2025) and can be accessed freely. The data are also available from the corresponding author on request.

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
