# Peer review of "Excitonic States in GaAs/AlxGa1−xAs Quantum Wells: Direct Coulomb Interaction Modeling via Finite Element Electrostatics and Parametric Analysis Under Impurity and Field Effects"

_nanomaterials, 2025, doi:10.3390/nano15171345_

Round 1

Reviewer 1 Report

Comments and Suggestions for Authors

Journal: Nanomaterials (ISSN 2079-4991)

Manuscript ID: nanomaterials-3823099

Type: Article

Title: Excitonic States in  Quantum Wells: Direct Coulomb Interaction Modeling via Finite Element Electrostatics and Parametric Analysis under Impurity and Field Effects

Authors: Fabian Andres Castaño , David Laroze , carlos alberto duque echeverri *

Section

Theory and Simulation of Nanostructures

Special Issue

Theoretical Calculation Study of Nanomaterials: 2nd Edition

This paper, titled "Excitonic States in  A Quantum Wells: Direct Coulomb Interaction Modeling via Finite Element Electrostatics and Parametric Analysis Under Impurity and Field Effects", focuses on modeling the behavior of excitons (bound electron-hole pairs) in Gallium Arsenide (GaAs) quantum wells.

  1. The paper's novelty is described as the comparison of two independent computational methods: a numerical quadrature approach using elliptic function corrections and a fully self-consistent electrostatic formulation using the FEM electrostatics module in COMSOL-Multiphysics. What are the specific theoretical advantages and disadvantages of each method beyond the performance metrics presented in the results section?
  2. The well width of the GaAs QW is varied parametrically from 20 nm to 200 nm. How were the barrier widths and the overall simulation domain size determined to ensure that the infinity domain boundary conditions did not affect the results for the largest well widths?
  3. Figure 2 shows the electron and hole wavefunctions for a well width of 80 nm. This is a relatively wide well. Why were these specific parameters chosen for this illustrative figure? Would a narrower well (e.g., 20 nm) show a more dramatic effect of the impurity and fields?
  4. While the paper uses COMSOL-Multiphysics, the manuscript doesn't explicitly mention the specific modules or versions used. Providing this information would enhance reproducibility.
  5. The paper provides a GitHub link to the source code, which is excellent. However, a more detailed explanation of the code, including instructions on how to use it, would be beneficial for other researchers trying to replicate or extend this work.

Author Response

See the pdf file

Reviewer 2 Report

Comments and Suggestions for Authors

In this study, the authors investigate excitonic states in GaAs quantum wells with Alâ‚€.₃Gaâ‚€.₇As barriers. Particularly, the authors modeled electron and hole states by direct numerical solutions of the Schrödinger equation in cylindrical coordinates using the finite element method (FEM), avoiding the use of trial wavefunctions. Exciton binding energies are calculated using two independent approaches: a parallelized numerical integration scheme based on elliptic corrections and a finite element formulation implemented in COMSOL Multiphysics, which agree within 1%. The results show stronger binding in narrow wells, reduced by donor impurities and electric fields, and partially recovered by magnetic fields. The FEM–electrostatics method is presented as efficient and flexible for complex systems. While the study is well-designed, several issues should be addressed.

  1. The authors state that the Schrödinger equations for the electron and hole are solved independently, with the Coulomb term added later. This approach may underestimate the absolute binding energy. Please clarify how significant this underestimation might be compared to variational correlated two-particle approaches.
  2. The effective masses and band offsets are assumed constant (line 129, page 4) across all well widths and field strengths. In real systems, these can vary with strain, temperature, and composition. It would strengthen the work if the authors discuss the sensitivity of their results to small changes in these parameters.
  3. In Section 2.1 (L135–137) and Section 3.1 (L170–172), the impurity is modeled as a point Coulomb potential located at the center of the GaAs QW along the symmetry axis (ρ = 0). Please discuss how an off-axis or spatially distributed impurity would affect carrier localization and exciton binding energy, and whether the proposed FEM approach can handle such configurations.
  4. Several typographical errors should be corrected. For example, “donnor” should be replaced with “donor” for consistency.

Author Response

See the pdf file

Reviewer 3 Report

Comments and Suggestions for Authors

This manuscript presents a numerical investigation of excitonic states in GaAs/Al 0.3Ga0.7As quantum wells (QWs), examining the effects of impurities, electric/magnetic fields, and well widths on exciton binding energies. The authors employ two complementary methods—numerical integration with elliptic function corrections and finite element method (FEM) via COMSOL Multiphysics—to compute Coulomb interactions. The authors report good agreement (≤1%) between the two methods, explore donor/acceptor impurity and field effects, and claim advantages of the COMSOL electrostatic formulation in accuracy and speed. However, several aspects of the manuscript could benefit from revision to improve clarity and scientific impact. Below are some specific comments on the manuscript:

1. There is a spelling mistake in Figure 2. “donnor” should be “donor”.

2. Figure 2 has two “(e)” panels in the caption.

3. The wavefunction distribution graph (Figure 2) lacks a color scale, and the text in the potential field graph (Figure 8) is not clear enough.

4. The manuscript considers only single donor/acceptor impurities with fixed concentrations nimp = ±1. Actually, QWs often contain impurity distributions, and concentration variations could affect exciton localization. The authors should discuss why a single impurity model is sufficient. 

5. The author should specify the COMSOL version, module name, and eigenvalue solver name, which are critical for reproducing the results.

6. In introduction, the authors write: “Excitons in semiconductor quantum wells (QWs) remain a central topic in condensed matter physics...These bound electron-hole pairs exhibit enhanced binding energies in low-dimensional systems, and in GaAs/AlGaAs heterostructures in particular, such energies can signiffcantly exceed bulk values, enabling stable excitonic states even at room temperature [1–6]…” The general reference list in the introduction seems a bit thin, considering the evolution in the field within the recent years. To give the readers a much broader view, recent developments concerning on excitons in semiconductor QWs, such as Laser & Photonics Reviews 17(3), 2200455 (2023); DOI: 10.1364/OL.452477, etc. should be considered, so that the readers can be clear about the state-of-the-art of this topic.

Comments on the Quality of English Language

There is a spelling mistake in the manucript. For example, “donnor” should be “donor” in Figure 2(e, f).

Author Response

See the pdf file
